# Effect of Potato Vine and Leaf Mixed Silage Compared to Whole Corn Crops on Growth Performance, Apparent Digestibility, and Serum Biochemical Characteristics of Fattening Angus Bull

**DOI:** 10.3390/ani13142284

**Published:** 2023-07-12

**Authors:** Jiajie Deng, Siyu Zhang, Yingqi Li, Changxiao Shi, Xinjun Qiu, Binghai Cao, Yang He, Huawei Su

**Affiliations:** 1State Key Laboratory of Animal Nutrition, College of Animal Science and Technology, China Agricultural University, Beijing 100107, China; 18811560713@163.com (J.D.); s20213040648@cau.edu.cn (S.Z.); liyingqi1230@163.com (Y.L.); scx15646702767@163.com (C.S.); caobh@cau.edu.cn (B.C.); heycau@163.com (Y.H.); 2College of Animal Science and Technology, Hainan University, Haikou 570228, China; qiuxinjun@hainanu.edu.cn

**Keywords:** beef cattle, potato vine and leaf mixed silage, growth performance, nutrients metabolism

## Abstract

**Simple Summary:**

Potato (*Solanum tuberosum* L.) is a worldwide-used food and is a perennial plant in the nightshade family Solanaceae. Potato vine and leaf mixed silage (PVS) is detoxicated after ensiling and can be used as high-quality roughage for ruminant animals. Because the PVS’s inadequate nutrient content cannot meet the requirement of ensiling, especially due to its lack of the water-soluble carbohydrates, the only method for ensiling consists of mixing with different feed ingredients. In this trial, the blood urea nitrogen and crude protein digestibility of the Angus bull were significantly higher when fed with PVS. The result revealed a high content of crude protein, crude protein apparent digestibility, and ash content in PVS compared with CS in the feeding trial. PVS significantly decreased Angus bulls’ average daily gain and feed conversion rate and increased high-density lipoprotein cholesterol (HDL-C) and low-density lipoprotein cholesterol (LDL-C). Because potato vine and leaf as a by-product of potato production can achieve the same economic production targets for beef cattle and still meet the requirements of sustainable development, the aim of this trial was therefore to investigate the possibilities of PVS in ruminant silage substitution and hopefully to give some guidance on conventional silage substitution.

**Abstract:**

This study aims to explore the different growth performances of the Angus bull on potato vine and leaf mixed silage in the early fattening period and to provide a reference animal production trial. Thirty-six 13-month-old Angus bulls were divided into three groups with 403.22 ± 38.97 kg initial body weight and fed with three different silage diets: (1) control: whole-plant corn silage as control (CS); (2) treatment 1: 50% whole-plant corn +50% potato vine and leaf silage (PVS1); and (3) treatment 2: 75% potato vine and leaf +15% rice straw +10% cornmeal silage (PVS2). After the 14 days pre-feeding, the formal experiment was carried out for 89 days. The result showed that the ash content of the potato vine and leaf mixed silage (PVS) in the treatment groups was higher than that in control group, and the ash content of PVS1 and PVS2 even reached 10.42% and 18.48% (DM%), respectively, which was much higher than that of the CS group at 4.94%. The crude protein content in silage also increased with the additional amount of potato vine and leaf. The apparent crude protein digestibility of the PVS groups was also significantly higher than that of the CS group (*p* < 0.05). In terms of serum biochemical indexes, blood urea nitrogen (BUN) in the experimental groups was significantly higher than in the control group (*p* < 0.05). Compared with PVS2, cholesterol (CHO) was significantly lower in the CS and PVS1 groups (*p* < 0.05). Moreover, the high-density lipoprotein cholesterol (HDL-C) and low-density lipoprotein cholesterol (LDL-C) of PVS2 were significantly higher than those of the CS and PVS1 group (*p* < 0.05), and daily gain (ADG) as a key production index had a significantly negative correlation with the CHO (r = −0.38, *p* < 0.05) and HDL-C (r = −0.40, *p* < 0.05) of cattle. In conclusion, PVS had higher crude protein content and ash but less starch than whole-corn silage. The PVS could replace whole-plant corn silage at the same dry matter status and did not affect the weight gain in this trial.

## 1. Introduction

Annual potato (*Solanum tuberosum* L.) production in China is increasing yearly. From FAO potato production data, China is the largest potato cultivation country, with an output of 94,300,000 tons and covering 5,780,000 hectares in 2021. For a long time, potato vine and leaf were either abandoned on farmland or burnt in China, releasing large amounts of greenhouse gases (GHG) [1]. The use of this agricultural by-product has become a major problem for agricultural production in recent years. As by-products, potato vine and leaf contain α-solanine and α-chaconine [2,3], which causes mammal solanine poisoning, such as gastrointestinal and nervous system disorders [4,5]. According to a previous research report, α-solanine and α-chaconine can be converted into aglyconesby micro-organisms during ensiling and rumen fermentation [6,7], which can protect flavonoids from complete degradation in the rumen [8]. Studies in dairy cows have shown that flavonoids increase whole-lactation milk yield [9] and reduce liver damage [10] in feeding trials. Additionally there is a meta-analysis showing that dietary supplementation with flavonoids can significantly increase daily weight gain (ADG) and backfat thickness (BFT) [11]. In Noordar’s research [12], the water-soluble carbohydrate (WSC) in potato vine and leaf was found to be 3.7–8.6% (DM), which was not sufficient to support the carbohydrate consumption during the whole fermentation process. However, in the research of Parfitt et al. [13], the crude protein of potato vine and leaf can reach about 11.4%. Therefore, mixing potato vine and leaf with whole-plant corn or adding corn meal is a good method to facilitate the fermentation process and improve mixed-silage quality. These supplements will not only improve the quality of the mixed silage but also improve cattle’s palatability and feed intake [14].

In the FAO’s 2013 investigation, ruminant GHG emissions represented 81% of total animal production GHG emissions [15]. For the GHG effect capacity, methane is 28 times greater relative to CO_2_ [16]. Meanwhile, in Guo et al.’s in vitro ruminal gas production experiment, the methane production of potato vine and leaf was significantly lower than that of whole-plant corn silage (CS) [17]. Thus, potato vine and leaf mixed silage (PVS) may be a potential solution for reducing GHG emissions.

Although PVS can reduce environmental pollution in ruminant production by mitigating the methane emission, research on PVS’s effect on ruminant production performance, particularly in beef cattle roughage utilization, is required to evaluate whether PVS can be a replacement of CS. Moreover, PVS differs from whole-plant corn silage in terms of nutrient composition, which alters the digestion and absorption of nutrients and enhances the growth performance of beef cattle. However, in Guo et al.’s [17] research, adding rice bran and whole-plant corn significantly improved the PVS quality by decreasing the moisture and increasing WSC.

For the utilization of plant leaf and potato, the main research on silage substitution is now in sweet potato vine silage, while for other processed potato waste, potato pulp silage is also used as a corn silage substitute for beef cattle fattening. Feed trials on Japanese black (Wagyu) mature cull cows showed that potato pulp silage had no effect on cattle growth performance [18], and for other small ruminant feeding trials, there was no adverse effect of replacing alfalfa hay with potato vine hay or potato vine silage on fattening lambs [19]. Nearly all the PVS research has mainly focused on silage nutrients and fermentation quality. Therefore, in this study, we performed feeding trials to investigate growth performance, serum biochemical indexes, and apparent nutrient digestibility of Angus beef cattle fed with mixed silage during the fattening period. Here, we compared the growth performance, apparent digestibility, and biochemical indexes of the Angus beef cattle fed with diets in different silage compositions. We hypothesized that different silages in the diet may affect the production of beef cattle due to biochemical indexes, especially in crude protein digestibility and blood urea nitrogen of Angus beef cattle.

## 2. Materials and Methods

### 2.1. Ensiling of Three Silage Raw Materials

The whole-plant corn and potato vine and leaf were harvested and then processed at Benwang Farm (106°028′ E 38°202′ N, Ningxia, China), a cooperative farm with China Agricultural University, from 5–15 September 2019. The average daily minimum and maximum ambient air temperature from August to December 2020 during the feeding trial was 6.0–20.3 °C, and the average relative humidity was 50.2% (recorded by China Meteorological Administration).

Whole-plant corn was harvested at the half to two-thirds milk-line stage and chopped with a silage harvester to a consistent length of 19 mm, sealed with oxygen barrier film (Shanghai Comiy BioTechnology LTC., Shanghai, China), and stored in a silage pit with a density near 700 kg/m^3^. Potato vine and leaf was harvested when most of the potato vine and leaf turned from green to yellow and withered, using a potato harvester to separate the potato fruit from the potato vine and leaf; we then collected the vine and leaf and wrapped silages with a radius of 40 cm, a height of 80 cm, and a density of nearly 800 kg/m^3^. After ensiling, we checked weekly for breaks and leaks.

### 2.2. Animal Diets and Management

Three silage treatments were used: (1) whole-plant corn silage (CON); (2) potato vine and leaf mixed silage 1 (PVS1): 50% potato vine and leaf + 50% whole corn; and (3) potato vine and leaf mixed silage 2 (PVS2): 75% potato vine and leaf + 15% rice straw + 10% corn meal. The three silages were prepared following the same silage-management methods. The corn silage insulation film was opened after 60 d of continuous fermentation, and the wrapped silage was opened during the feeding experiment from 20 August 2020 to 4 December 2020. The three silages were sampled on the top, middle, and bottom layers for further nutrient analysis, which can be seen in Table 1.

Thirty-six 13-month-old Angus bulls were divided into 9 pens according to their weight of 403.22 ± 38.97 kg (Mean ± SD). The cattle were randomly allocated in 9 pens with no statistical difference in initial weight fed with the three silages diets, and cattle in pens 1–3 were fed a diet containing CS, pens 4–6 a diet containing PVS1, and pens 7–9 a diet containing PVS2. The feeding frequency of TMR was twice a day at 07:00 h and 16:00 h (GMT + 8) with ad libitum drink. Restricted feeding was applied in this feeding trial to control the same dry matter intake of different silages. All the cattle in our trial were compliant with the Guidelines of the Animal Care Committee and animal welfare guidelines of China Agricultural University (AW82303202-1-1).

To guarantee the same dry matter intake of the three diets, the dry matter of the three silages was analyzed before the feeding trial (Dry Matter%; CS:30.07, PVS1:25.59, and PVS2:35.71). The three feeding test groups were designed to ensure that the dry matter intake was the same. The diet groups and composition are shown in Table 2.

### 2.3. Sample Collection and Analysis

All bulls were only weighed in the morning at day 0 and day 89 to minimize the stress during the feeding trial. Average daily gain (ADG) was calculated by subtracting the initial body weight from the final body weight, then dividing it by the duration of the feeding trial over 89 days. Then, the feed conversion ratio (FCR) was calculated by dividing the dry matter intake (DMI) by the weight gained during the experiment.

From day 87 to day 88 in the morning, after 12 h of fasting and before morning feeding, blood was collected from the caudal roots of individual bulls using vacuum tubes with a blood collection needle. After standing for 30 min, samples were centrifuged at 3500 r/min for 10 min, and then, the serum was collected by pipette. The collected serum was kept in a refrigerator at −80 °C for further biochemical indexes analysis. The biochemical parameters of the serum samples were measured by an automatic hematology analyzer (Hitachi 7020, Tokyo, Japan) with the biochemical kits: alanine aminotransferase (ALT), aspartate aminotransferase (AST), total protein (TP), alkaline phosphatase (ALP), cholesterol (CHO), triglyceride (TG), high-density lipoprotein cholesterol (HDL-C), low-density lipoprotein cholesterol (LDL-C), non-esterified fatty acid (NEFA), beta-hydroxybutyrate (BHB), blood urea nitrogen (BUN), glucose (GLU), creatinine (CREA), and albumin (ALB) concentrations were determined using commercial test kits (Beijing Jiuqiang Bio-Technique Co., Ltd., Beijing, China). Antioxidant indexes, superoxide dismutase (SOD), malondialdehyde (MDA), glutathione peroxidase (GSH-PX), catalase (CAT), total antioxidant capacity (T-AOC), reactive oxygen species (ROS), and oxidative stress index (OSI) were also analyzed with the commercial test kits (Nanjing Jiancheng Biological Engineering Co., Ltd., Nanjing, China).

The acid detergent fiber (ADF) and neutral detergent fiber (NDF) of the feeds were determined by the ANKOM 2000 fiber analyzer (ANKOM Technologies, Macedon, NY, USA). Heat-stable alpha-amylase was also used in the NDF determination according to the method of Van Soest [20]. The DM (Method 934.01), ether extract (EE; Method 920.39), and crude protein (CP; Method 990.03) of the feed were determined according to the Association of Official Analytical Chemists (AOAC, 2000). Acid-insoluble ash (AIA) was analyzed followed by the method of Van Keulen [21].

The apparent digestibility of the nutrients (ADN) was calculated with the following formula:(1)$$\mathrm{ADN}\%=\left[1-\frac{\left(\%AIA_{Diet}\times \%{Nut}_{Feces}\right)}{\left(\%AIA_{Feces}\times \%{Nut}_{Diet}\right)}\right]\times 100\%$$

*AIA_Diet_* is the acid-insoluble ash content in the diet, *Nut_Feces_* is the content of that nutrient in the feces, *AIA_Feces_* is the acid-insoluble ash content in the feces, and *Nut_Diet_* is the content of that nutrient in the diet.

### 2.4. Statistical Analysis

All data were initially processed in Excel 2019 (Microsoft Office, Washington, DC, USA) and then imported into SPSS 26.0 (IBM, Armonk, NY, USA) using a general linear model (GLM) for the following model. Y_ij_ = μ + τi + ϵij, where Y_ij_ is the dependent variable, μ is the common effect of the whole experiment, τ_i_ represents the i_th_ diet effect, and ϵ_ij_ represents the random error present in the j_th_ observation point of the i_th_ diet. Spearman correlation heatmap was analyzed and drawn using R Statistical Software (v4.1.2, R Core Team 2021, Vienna, Austria).

Results are expressed as mean with standard error (SEM). Post hoc was calculated using the last significant differences (LSD). When the *p*-value is less than 0.05, there is a significant difference between different groups marked with different lower-case letters within the same row in tables.

## 3. Results

### 3.1. Growth Performance of Cattle

There was no significant difference in body weight between the groups at the start of the trial (*p >* 0.05), and there was potential for the final weight in the CS group to be higher than the PVS1 and PVS2 groups in Table 3. The three silage diets had no significant difference in dry matter intake (DMI) during the feeding trial. However, the CS group was significantly lower than the PVS1 and PVS2 groups in feed conversion ratio and higher in daily weight gain (*p* < 0.05).

### 3.2. Dietary Nutrient Composition and Apparent Digestibility

The impact on apparent digestibility of nutrients by adding different silage types to the diet is shown in Table 4, which shows that the apparent digestibility of crude protein was significantly higher in the PVS group than in the CS group (*p* < 0.05).

### 3.3. Biochemical Routine Indexes and Antioxidant Indexes of Beef Cattle Serum

The effects of adding different silage types to the diet on blood metabolic indexes are shown in Table 5. Among the blood biochemical indexes, the lipid metabolic indexes of the PVS2 group were significantly different from those of the CS and PVS1 groups. Total cholesterol (CHO) was significantly higher (*p* < 0.05) relative to CS and PVS1. High-density lipoprotein cholesterol (HDL-C) and low-density lipoprotein cholesterol (LDL-C) were significantly higher than those of CS and PVS1 (*p* < 0.05), and free fatty acids (NEFA) were significantly higher in the PVS2 than in the CS and PVS1 groups (*p* < 0.05). The different silage diets affect lipid metabolism and blood urea nitrogen in the blood. Figure 1 shows the effect of different silages on the serum antioxidant indexes of Angus bulls, and there was also no significant difference in serum antioxidant indexes in the beef cattle fed different diets among the three groups. Figure 2 shows the Spearman correlation, and in the production indexes, ADG had a significant negative correlation with CHO (r = −0.38, *p* < 0.05) and HDL-C (r = −0.40, *p* < 0.05).

## 4. Discussion

### 4.1. PVS Diets Nutritional Properties and Effects of PVS on the Growth Performance of Fattening Angus Bulls

The diet composition of the PVS group contained a significantly higher content of crude protein than the CS group, which is similar to the findings of Parfitt et al. [13], who measured the nutrients of potato vine and leaf and obtained a CP of about 11.4% in dry matter base. Compared to whole-plant corn silage, which contains approximately 6.5–8.5% CP in dry matter [22], this is consistent with the result shown in Table 1.

At the no significant initial weight and DMI status, the beef cattle in the PVS test group had a significantly lower ADG and FCR than the control group. Although the PVS1 group was significantly higher than the CS and PVS2 groups in DM digestibility, the CS group was significantly higher in ADG, which may probably be due to the PVS groups having lower EE and ME in the diet. However, in Malecky et al.’s [19] research, replacing 50% alfalfa hay with potato vine and leaf silage increased the ADG and reduced the FCR of sheep. Though the ADG of the CS group was significantly higher than the other two groups, the final weight of the three groups of cattle had no significant difference. The reason for the non-significant final weight may be due to the higher CP content in the PVS diets groups. The different nutrient intake of growing beef cattle, such as different CP and ether extract, may affect their metabolic and weight gain requirements [23,24]. Research in Brahman crossbred and Holstein-Friesian breeds found that during energy restriction, a higher CP intake in the diet increased the IGF-1 level in plasma, which resulted in a higher liveweight gain [25].

The CP content of PVS1 and PVS2 was higher than CS, and CP content increased as the proportion of potato vine and leaf in the silage increased. The CP content of PVS is consistent with the results of Salehi et al. [26], as the CP content in potato vine and leaf ranged from 13–17% due to different treatments. Moreover, there was no significant difference in NDF of the three diets, as NDF is positively related to DMI [27], which is consistent with the lack of difference among the DMI of the three groups of cattle. However, the starch of the CS diet is higher, which shows that the cattle fed with PVS diets had a lower energy intake. The content of starch and NDF can affect the rumination and rumen pH, as lower NDF-content diets postpone the daily rumination rhythm and acrophase [28,29]. Higher ash in the diet usually means a lower relative feed value (RFV), and higher ash content in diets can also lead to digestive problems, including reduced feed intake and poor nutrient absorption. The production indexes with significantly lower ADG and higher FCR in the PVS groups are also due to the higher ash content in diets. Therefore, the potato-vine- and leaf-harvesting process also needs to be optimized so that the soil carried by the underground fruit is not carried into the silage.

The amount of digestible CP that is required by cattle depends on various factors, such as their age, weight, and production stage. In this trial, the PVS groups had a significant increase in CP digestibility. This is consistent with research on growing Charolais bulls, which indicate that a high-protein diet brought a higher CP digestibility [30]. In addition, research conducted on tropical cattle showed that adding a high content of crude CP-supplementing sweet potato vine silage can improve feed digestibility [31]. The explanation for this phenomenon is that bulls receiving a high-CP diet had higher ruminal bacterial protein levels in PVS than CS [32]. Furthermore, many different factors are responsible for CP digestibility, such as protein source, dietary CP level, animal condition, and breed [33].

### 4.2. Effect of PVS on Blood Biochemical Indicators of Angus Beef Cattle

Blood biochemical indicators are essential to show the metabolic levels of beef cattle. Among the blood biochemical indicators, serum alanine aminotransferase (ALT) and aspartate transferase (AST) are the active enzymes that reflect liver function [34], with a normal range of ALT in cattle blood ranging from 11–40 U/L [35]. In this trial, ALT was detected at around 20 U/L and was not significantly different, demonstrating that its levels were normal in beef cattle. However, the AST level in PVS1 was above the warning value of 98 U/L [36], and this indicates that the PVS1 group cattle had liver dysfunction or damage [37]. Blood glucose is an important source of energy in beef cattle, and monitoring blood glucose, total protein (TP), and albumin (ALB) levels in cattle can be helpful in diagnosing cattle physiological and pathological states and managing various diseases and conditions [38,39,40]. In this experiment, there is no difference between Glu, TP, and ALB in CS and PVS groups, and this result is consistent with Xia et al.’s [32] research in Holstein bulls fed diets with different CP levels.

Blood urea nitrogen (BUN) is used as an indicator of protein metabolism and nutritional status in beef cattle [41]. In the present study, it was concluded that the ratio of BUN to CP in the diet was linearly related in cows [42,43,44], which is consistent with the results of this test. When there is an imbalance in the amino acid composition of beef cattle, the protein is less efficiently utilized and is converted into blood urea nitrogen [45,46]. Furthermore, lower NH3-N levels in the PVS group’s rumen liquid indicate that the efficiency of microbial protein synthesis in the PVS group is lower than in the CS group. Although the apparent digestibility of CP in the PVS diet is higher, the unreasonable amino acid composition ratio in it will lead to poor amino acid metabolic utilization in beef cattle [47]. This is consistent with the outcome of the ADG and FCR of the PVS group cattle, as the amino acid composition may not meet the essential amino acids requirement, and this is due to a rumen ammonia concentration above what is typically provided by a 12–13% protein diet that will not facilitate the enhanced utilization of ammonia nitrogen for microbial protein synthesis [48]. More research about the amino acid composition of PVS should be carried out, and diet supplement of amino acids is needed for the PVS diet.

As for indicators of lipid metabolism, lipids are an essential energy source for cattle and are also involved in many physiological functions, such as cell membrane structure and function, hormone synthesis, and immune function. The lipid metabolism indexes of the PVS2 group were different from those of the other groups, as reflected by the fact that CHO, NEFA, HDL-C, and LDL-C were higher than the other two groups. Higher CHO in the PVS2 group is possibly due to lower levels of lipid metabolism in the silage and the lower ether extract content [49]. The level of NEFA reflected the state of energy of the animal, with high concentrations indicating a negative energy balance in dairy cows [50], and the lower ME in the PVS2 group is consistent with the higher NEFA in the PVS2 group. In this study, the ADG of the cattle had a negative correlation with CHO and HDL-C, and cattle fed PVS had a higher CHO and HDL-C, which revealed that PVS2-diet cattle had hypothyroidism [51]. As the PVS2 group’s diet had a more proportionate amount of potato vine and leaf in the diet, the above results indicate that potato vine and leaf has an unreasonable fatty acids composition.

Excessive free radical production and redox destabilization in animals can be caused by environmental, management, nutritional, and internal stresses [52,53], which reflect the adverse effects of oxidative stress on the animal immune system and animal health [54]. In this feeding trial, the environment did not cause any heat stress. The level of free radicals represented by reactive oxygen species (ROS) in the blood and other biological antioxidants such as superoxide dismutase (SOD) and glutathione peroxidase (GSH-Px), etc., are indicators to evaluate oxidative stress [52]. However, in this feeding trial, there were no significant differences in blood antioxidant indicators in beef cattle fed all three diets, which indicated that the PVS did not have a side effect on the cattle’s oxidative stress compared with CS.

## 5. Conclusions

In this feeding trial, we evaluated the growth performance, apparent digestibility, and biochemical indexes in beef cattle fed with different potato vine and leaf mixed silage compared to the whole-crop corn silage. Regardless of how much whole-plant corn is replaced by potato vine and leaf, this will affect the apparent digestibility of crude protein. Combined with differences in PVS energy levels, these factors of PVS affect production indicators such as the daily weight gain and feed conversion rate. Further research should focus on the difference in amino acids and fatty acids composition between potato vine and leaf and whole-plant corn and their potential impact on animal production and meat quality.

## Figures and Tables

**Figure 1 animals-13-02284-f001:**
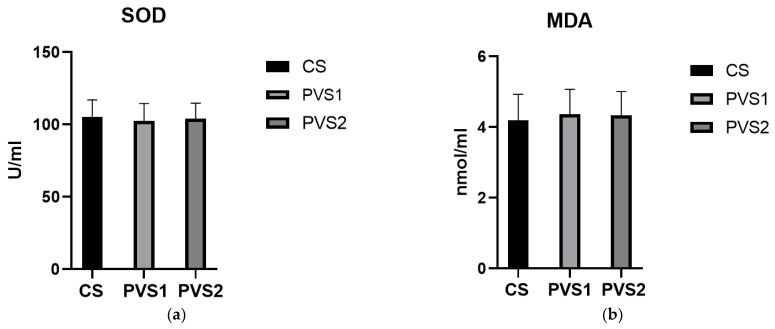
Serum antioxidant indexes of Angus beef cattle under three different silages. (**a**) This figure shows the superoxide dismutase value in the beef cattle serum of three groups (CS, corn silage; PVS1, potato vine and leaf mixed silage 1; PVS2, potato vine and leaf mixed silage 2). (**b**) This figure shows the malondialdehyde value in the beef cattle serum of three groups. (**c**) This figure shows the glutathione peroxidase value in the beef cattle serum of three groups. (**d**) This figure shows the catalase value in the beef cattle serum of three groups. (**e**) This figure shows the total antioxidant capacity value in the beef cattle serum of three groups. (**f**) This figure shows the reactive oxygen species value in the beef cattle serum of three groups. (**g**) This figure shows the oxidative stress index value in the beef cattle serum of three groups. SOD, superoxide dismutase, U/mL; MDA, malondialdehyde, nmol /mL; GSH-PX, glutathione peroxidase, U/mL; CAT, catalase, U/mL; T-AOC, total antioxidant capacity, U/mL; ROS, reactive oxygen species, U/mL; OSI, oxidative stress index, OSI = ROS/T-AOC. The bar chart shows the mean and SEM for each indicator; the error line is the positive standard error value.

**Figure 2 animals-13-02284-f002:**
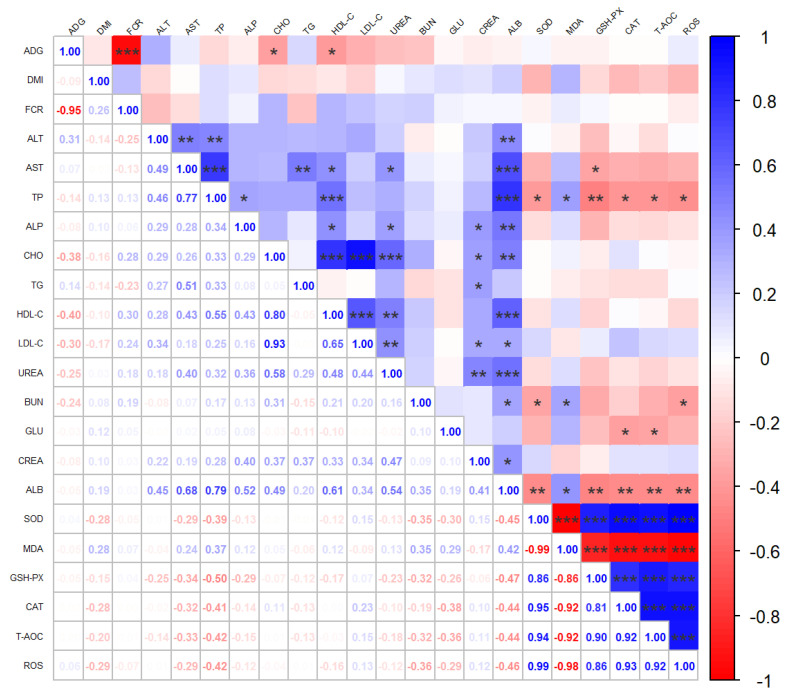
Heatmaps of Spearman correlation analysis. The correlation heatmap is used to represent significant statistical correlation values (r) between growth performance and biochemical indexes. In the heatmap, blue squares indicate significant positive correlations (r > 0.5, *p* < 0.05), white squares indicate non-significant correlations (*p* > 0.05), and red squares indicate significant negative correlations (r < −0.5, *p* < 0.05). * *p* < 0.05, ** *p* < 0.01, and *** *p* < 0.001.

**Table 1 animals-13-02284-t001:** Chemical composition of different silages.

Item	Treatment		
	CS ^1^	PVS1	PVS2
pH value	4.48	4.69	4.87
Nutritive values, %DM ^2^			
Dry matter	30.07	25.59	35.71
CP	6.91	7.84	8.45
EE	2.46	3.30	3.09
NDF	44.04	43.42	41.79
ADF	24.45	30.00	27.48
Ash	4.94	10.42	18.48
Starch	31.81	19.95	24.15
Metabolic energy ^3^, MJ/kg	11.08	10.51	9.38

^1^ CS, corn silage; PVS1, potato vine and leaf silage 1; PVS2, potato vine and leaf silage 2; ^2^ CP, crude protein; EE, ether extract; NDF, neutral detergent fiber; ADF, acid detergent fiber. ^3^ Metabolic energy (ME) was based on the total digestible nutrients of each feed ingredient. (ME = 0.82 × 4.409 × TDN, TDN (%) = 0.98 × (100-NDF_n_-CP-Ash-EE + IADICP) +k_dCP_×CP + 2.25 × (EE-1) + 0.75 × (NDF_n_−ADL) × [1−(ADL/(NDF_n_))^0.667^]−7), NDF_n_ = NDF-NDICP + IADICP, IADICP = 0.070 × ADICP K_dCP_ = exp(−0.0012 × ADICP), referring to NRBC (2016).

**Table 2 animals-13-02284-t002:** Composition, proportions, and nutritional values of the three diets.

Item%	Treatment		
	CS	PVS1	PVS2
Flaked corn	21.85	21.59	22.08
Soybean meal	5.59	5.36	5.79
Bran	3.73	3.57	3.86
Cotton meal	2.98	2.86	3.09
5% Premix ^1^	2.98	2.86	3.09
Dried distiller’s grains with soluble (DDGS)	2.98	2.86	3.09
Rapeseed meal	2.01	1.93	2.08
Probiotics ^2^	1.34	1.29	1.39
Fermented feed ^3^	4.97	4.76	5.14
Straw	2.48	2.38	2.57
Rice stalks	6.21	5.95	6.43
Silage	42.89	45.25	40.83
Nutritive values, % DM			
Dry matter	57.2	54.4	62.5
Crude protein	11.24	13.17	13.6
Ether extract	7.87	5.75	5.53
Neutral detergent fiber	28.35	28.76	27.7
Acid detergent fiber	17.00	17.00	15.41
Ash	8.51	9.11	10.81
Ca ^4^, %	0.41	0.39	0.42
P ^4^, %	0.15	0.14	0.15
Starch	31.81	19.95	24.57
Metabolic energy, MJ/kg ^5^	12.17	11.83	11.37

^1^ 5% Premix composition per kg vitamin A 120,000–200,000 IU, vitamin E ≥ 550 IU, D-biotin ≥ 0.3 mg, copper 0.16–0.5 g, manganese 0.6–2.4 g, selenium 1.6–10 mg, calcium 10.0–20.0%, vitamin D3 15,000–60,000 IU, nicotinamide ≥ 350 mg, iron 0.8–8.4 g, zinc 1.5–3.0 g, iodine 4–20 mg, sodium chloride 10.0–20.0%, and total phosphorus ≥ 2.0%. ^2^ Probiotics are Stirling S-7001 (Guangdong VTR Bio-Tech Co, Ltd., Zhuhai, China). ^3^ Fermented feed is yeast culture (Jiangsu Yiyuantai Bio-Tech Co., Ltd., Taixing, China). ^4^ Calcium and phosphorus are estimates, not actual analysis values (Database of Feed Composition and Nutritive Values in China, Version 31 2020, Beijing, China). ^5^ ME was based on the total digestible nutrients of each feed ingredient. (ME = 0.82 × 4.409 × TDN, TDN (%) = 0.98 × (100-NDF_n_-CP-Ash-EE + IADICP) + k_dCP_ × CP + 2.25 × (EE-1) + 0.75 × (NDF_n_-ADL) × [1−(ADL/(NDF_n_))^0.667^]−7), NDF_n_ = NDF-NDICP + IADICP, IADICP = 0.070 × ADICP K_dCP_ = exp(−0.0012 × ADICP), referring to NRBC (2016).

**Table 3 animals-13-02284-t003:** Growth performance of Angus beef cattle under three different silages.

	CS	PVS1	PVS2	SEM	*p*-Value
Initial weight, kg	400.36	400.27	404.73	7.06	0.960
Final weight, kg	528.82	500.82	506.00	8.69	0.388
ADG, kg/day	1.44 ^a^	1.13 ^b^	1.14 ^b^	0.52	0.014
DMI, kg/day	9.09	9.43	9.07	0.10	0.296
FCR	6.59 ^b^	8.64 ^a^	8.38 ^a^	0.33	0.022

Values with different superscripts within each row are significantly different (*p* < 0.05). DMI, dry matter intake; ADG, average daily gain; FCR, feed conversion ratio (feed/gain); SEM, standard error of the mean. CS, corn silage; PVS1, potato vine and leaf mixed silage 1; PVS2, potato vine and leaf mixed silage 2.

**Table 4 animals-13-02284-t004:** Apparent digestibility in Angus beef cattle under three different silages.

	CS	PVS1	PVS2	SEM	*p*-Value
Apparent digestibility%					
DM, %	69.55	70.53	69.32	0.35	0.377
CP, %	53.48 ^b^	70.18 ^a^	71.41 ^a^	3.47	0.028
EE, %	68.09	62.26	61.57	1.78	0.294
NDF, %	49.75	60.68	59.18	2.73	0.225
ADF, %	45.50	53.56	50.40	2.24	0.383

Values with different superscripts within each row are significantly different (*p* < 0.05). DM, dry matter; CP, crude protein; EE, ether extract; NDF, neutral detergent fiber; ADF, acid detergent fiber; CS, corn silage; PVS1, potato vine and leaf mixed silage 1; PVS2, potato vine and leaf mixed silage 2.

**Table 5 animals-13-02284-t005:** Serum metabolic indexes of Angus beef cattle under three different silages.

	CS	PVS1	PVS2	SEM	*p*-Value
Biochemical Index					
ALT	25.87	23.28	25.16	1.09	0.619
AST	90.64	103.05	94.75	3.87	0.423
TP	54.93	58.06	56.80	1.29	0.623
ALP	144.24	167.43	206.73	14.25	0.198
CHO	3.18 ^b^	3.54 ^b^	4.30 ^a^	0.12	<0.001
TG	0.20	0.22	0.18	0.01	0.085
HDL-C	0.97 ^c^	1.01 ^b^	1.19 ^a^	0.03	0.010
LDL-C	0.90 ^b^	0.96 ^b^	1.13 ^a^	0.03	0.003
BUN	2.88 ^b^	4.36 ^a^	4.31 ^a^	0.16	<0.001
NEFA	0.23 ^b^	0.29 ^a^	0.36 ^a^	0.07	0.036
GLU	1.79	1.61	1.66	0.44	0.602
CREA	87.83	93.33	92.50	2.55	0.652
ALB	29.46	31.50	32.13	0.58	0.147

Values with different superscripts within each row are significantly different (*p* < 0.05). ALT, alanine aminotransferase, U/L; AST, aspartate aminotransferase, U/L; TP, total protein, g /L; ALP, alkaline phosphatase, U/L; CHO, cholesterol, mmol/L; TG, triglyceride, mmol/L; HDL-C, high-density lipoprotein cholesterol, mmol/L; LDL-C, low-density lipoprotein cholesterol, mmol/L; NEFA, free fatty acid, mmol/L; BUN, blood urea nitrogen, mmol/L; GLU, glucose, mmol/L; CREA, creatinine, μmol/L; ALB, albumin, G /L. CS, corn silage; PVS1, potato vine and leaf mixed silage 1; PVS2, potato vine and leaf mixed silage 2.

## Data Availability

All the data are presented in the text and tables of this manuscript.

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
