# Peer review of "Effect of Potato Vine and Leaf Mixed Silage Compared to Whole Corn Crops on Growth Performance, Apparent Digestibility, and Serum Biochemical Characteristics of Fattening Angus Bull"

_animals, 2023, doi:10.3390/ani13142284_

Round 1

Reviewer 1 Report

The following recommendations are made to the authors

In the materials and methods, it should better describe how the experiment was carried out.

How was the daily feeding of the bulls?

How many times a day?

How often were the rejections withdrawn?

How was the drinking water?

what were the weather conditions?

In lines 163 and 164 use a formula editor

The authors should improve the discussion, since in most of it they only compare results with those already published in other manuscripts.

Further observations are indicated in the manuscript.

The writing must be improved and reviewed by a native English speaker

The writing must be improved and reviewed by a native English speaker

Author Response

Really grateful for your suggestion in the reviewed PDF file, and all the comments were carefully considered in attached word file. And revision part is highlighted. 

Reviewer 2 Report

Research concerns an important and novel by product utilisation issue of the beef production (feeding, afttening). The article is clear, and interesting.

In my opinion, the manuscript could be suitable for publication after revision, addressing the following comments:

Please check the color, type and size of the font used.

-          in line 26 and line 106 red color

-          the line spacing: in lines 28-31.

-          in line 119 change  table to Table

-in line 459 dairy to Dairy

Please check guidelines for references section.

Please insert further literature data such as:

Ali, A.I.M.; Wassie, S.E.; Korir, D.; Merbold, L.; Goopy, J.P.; Butterbach-Bahl, K.; Dickhoefer, U.; Schlecht, E. Supplementing Tropical Cattle for Improved Nutrient Utilization and Reduced Enteric Methane Emissions. Animals 2019, 9, 210. https://doi.org/10.3390/ani9050210

 Please give us more information about methods including serum examination. Please define kit names, which were used for detection of antioxidant satus of young fattening bulls.

Results Discussion

Please clarify the relationship (coefficient of correlation) between serum parameters, antioxidant indices with performance results.

The conclusions are correct and based on the results that are provided.

Author Response

Really grateful for your suggestion, and all the comments were carefully considered in attached word file. And revision part is highlighted.

Reviewer 3 Report

Abstract

Must be corrected there is no need for space between treatments

Introduction

The hypothesis of the work must be clearly presented

M&M's

Clearly present the experimental design of the study

Line 163-164: These lines should be formatted as the rest of the text

Discussion

I recommend that the discussion not be done on topics such as the current form, as it is understood that there is no relationship between the variables studied, since the blood parameters can help explain the performance and digestibility results

Conclusion.

It must be rewritten and present the main results of the study. In its present form it still presents part of the discussion in it.

Author Response

Really grateful for your suggestion, and all the comments were carefully considered in the attached word file. And revision part is highlighted.

Round 2

Reviewer 3 Report

no comment